METHODS

# ConNIS and labeling instability: New statistical methods for improving the detection of essential genes in TraDIS libraries

Moritz Hanke[1]*, Theresa Harten[2], Ronja Foraita[1]

**1** Department Statistical Methods in Epidemiology, Leibniz Institute for Prevention Research and Epidemiology – BIPS, Bremen, Bremen, Germany, **2** Independent Researcher, Hamburg, Germany

\* hanke@leibniz-bips.de

## Abstract

The identification of essential genes in *Transposon Directed Insertion Site Sequencing (TraDIS)* data relies on the assumption that transposon insertions occur randomly in non-essential regions, leaving essential genes largely insertion-free. While intragenic insertion-free sequences have been considered as a reliable indicator for gene essentiality, so far, no exact probability distribution for these sequences has been proposed. Further, many methods require setting thresholds or parameter values *a priori* without providing any statistical basis, limiting the comparability of results. Here, we introduce *Consecutive Non-Insertion Sites* (*ConNIS*), a novel method for gene essentiality determination. *ConNIS* provides an analytic solution for the probability of observing insertion-free sequences within genes of given length and considers variation in insertion density across the genome. Based on an extensive simulation study and different real-world scenarios, *ConNIS* was found to be superior to prevalent state-of-the-art methods, particularly when libraries had only a low or medium insertion density. In addition, our results showed that the precision of existing methods can be improved by incorporating a simple weighting factor for the genome-wide insertion density. To set methodically embedded parameter and threshold values of *TraDIS* methods a subsample-based instability criterion was developed. Application of this criterion in real and synthetic data settings demonstrated its effectiveness in selecting well-suited parameter/threshold values across methods. An R package and an interactive web application are provided to facilitate application and reproducibility.

## Author summary

Identifying essential genes in bacteria is key to understanding their ability to survive, which can, for example, be applied to the development of new treatments. One way to do identify these genes is by creating libraries where small DNA fragments ("insertions") are randomly placed in the genome: essential genes

**Data availability statement:** All data and code used for running experiments, model fitting, and plotting is available on a GitHub repository at https://github.com/bips-hb/ConNIS_results. We have also used Zenodo to assign a DOI to the repository in a zip format: https://doi.org/10.5281/zenodo.16790977 Additional real-world results are available under https://zenodo.org/records/18538450. All results can be interactively explored under https://connis.bips.eu. The new methods are made available as R package under https://github.com/bips-hb/ConNIS.

**Funding:** The author(s) received no specific funding for this work.

**Competing interests:** The authors have declared that no competing interests exist.

tend to remain insertion-free because insertions disrupt their function. The challenge is to determine whether a (long) uninterrupted sequence is due to chance or because the gene is truly essential. Here, we present *Consecutive Non-Insertion Sites (ConNIS)*, a statistical method that calculates the probability of such insertion-free sequences. Extensive comparisons on simulated and real datasets show that *ConNIS* outperforms existing methods, especially when a library is rather sparse in terms of the total number of insertion sites. Since many analysis methods rely on parameter values that have to be set before the analysis and can heavily influence the final results, we also propose a data-driven approach to set these values, making results more comparable across studies. Our methods are freely available as an R package and all results are presented in a web app.

## Introduction

Determination of genes essential for the growth and survival of bacteria has been of major interest in genetic research as it provides a deeper understanding of lifestyle and adaptation [1–3]. While site-directed mutagenesis approaches determine essential genes accurately, such methods are laborious and time-consuming when performed globally. Consequently, whole genome analyses have only been conducted for well-known model organisms such as *Escherichia coli*, i.e., the Keio library [4]. In the last decade, wider availability of high-throughput sequencing methods initiated a shift from single-gene to whole-genome analysis, resulting in the development of *transposon insertion sequencing (TIS)* methods. Employing transposons that are randomly inserted into the genome enables researchers to generate large mutant libraries and to characterize them by the location of insertion sites (IS) via high-throughput sequencing. *Transposon directed insertion site sequencing (TraDIS)* is a widely applied *TIS* method [5–8] and has been established for the determination of essential genes in various scientific set-ups [9–15].

A key challenge in *TIS* studies resides in the statistical analysis, which typically aims to maximize the detection of true positives (essential genes) while minimizing the number of false positives (non-essential genes incorrectly identified as 'essential'). Although there are multiple software suites and packages, not every method embedded therein will be equally suitable for the analysis of the obtained data set. For example, sliding window approaches [16,17] and Hidden Markov Models [18,19] have been proposed for the analysis of high-density libraries which regularly originate from *mariner*-transposon-based mutagenesis [20,21]. However, many *TIS* studies utilize *Tn5*-based transposons, which have different underlying assumptions and constraints in terms of the data generating process: Unlike *mariner* transposons, *Tn5* insertions do not depend on the presence of specific motifs and can theoretically occur in any non-essential region of the genome [6,22]. Nevertheless, *Tn5*-based libraries reported so far tend to be less dense than *mariner*-based libraries. Consequently, observing larger genomic regions lacking IS just by chance becomes more likely in *Tn5*-based libraries. Furthermore, the set of detected IS across the genome

rarely displays a uniform distribution of gene-wise insertion densities. Reasons may be transposon-driven preferences for GC- or AT-rich regions [23–26] and genomic hot- or coldspots, i.e., genomic regions of notably higher or lower insertion densities, respectively [24,27–29].

So far, a couple of *Tn5*-based statistical methods for identifying essential genes have been proposed. Burger et al. [30] suggest estimating the probability of observing several IS within a gene of a given length based on a binomial distribution using the genome-wide insertion density as success probability. The *Tn5Gaps* method of the Transit package [31] uses a Gumbel distribution to approximate the probabilities of observed IS-free gaps along the genome. Essentiality is determined by the largest gap within or partially overlapping a gene. Since both methods rely on *p*-values derived from thousands of genes, the authors recommend correcting for multiple testing. However, they did not evaluate how different correction approaches might affect the identification of essential genes. Alternatively, the Bio-TraDIS software package [32] avoids the multiple testing problem by heuristically leveraging an often observed bimodal distribution of gene-wise insertion densities. Combining an exponential distribution (for essential genes) with a gamma distribution (for non-essential genes), genes are labeled as 'essential', 'non-essential', or 'ambiguous' based on an *a priori* set $\log_2$ likelihood ratio threshold. In practice, a clear distinction between the two distributions is not always guaranteed [6], and the threshold values are usually set arbitrarily. The recently proposed Bayesian method *InsDens* calculates the posterior probability of a gene being essential [33] and the authors suggest to use Bayesian decision theory to set a posterior probability threshold. Although this method offers a clear interpretation, it requires choosing an *a priori* probability distribution parameter, too, which can influence the outcome.

Some but rather limited comparative studies of *TIS* methods are available. Based on high-density library data, Larivière et al. [20] draw a comparison between the bimodal approach of Bio-TraDIS, the *Tn5Gaps* method and a custom modification of the bimodal approach [11]. However, only two threshold values were applied and no performance analysis under different controlled data-generating processes was described. While Nlebedim et al. [33] used four different parameter combinations to generate synthetic data, the only method applied in the analysis was their method *InsDens*. The additional analysis of three real-world datasets using Bio-TraDIS suffers from the application of only one and rather low threshold value. Similarly, Ghomi et al. [34] proposed to use the non-parametric clustering algorithm embedded in the DBSCAN R package for the sole reason that it allows omitting the heuristic setting of threshold values required by Bio-TraDIS. Again, only a single and rather low threshold value for comparison using Bio-TraDIS was applied. However, the DBSCAN clustering algorithm itself requires setting two *a priori* parameter values but the authors did not report how different values might affect labeling performance, nor did they provide guidance on how to choose appropriate values.

The widespread use of *TIS*, particularly *Tn5* statistical analysis methods, contrasts with the lack of systematic reviews of these methods, especially when considering different data-generating processes. Furthermore, a transparent, comprehensive statistical method for setting threshold or parameter values in *TIS* methods is missing. As a consequence, publications often only justify the choice of methods and parameters by citing prior studies that used similar approaches and parameters. In addition, most studies lack sensitivity analyses for their threshold and parameter values. A common practice is truncating the 5'- and/or 3' ends of genes by several base pairs or up to 20%, to align with the assumption that the gene ends are generally non-essential [6,13,35–38], yet this approach is never investigated in sensitivity analyses.

At this scientific stage, we provide the following contributions to the statistical detection of essential genes. First, we introduce *Consecutive Non-Insertion Sites (ConNIS)*, a novel method that determines gene essentiality based on insertion-free sequences within genes. *ConNIS* provides an analytical solution for the probability of observing the longest insertion-free sequence within a gene, based on its length and the number of IS under the assumption of being non-essential. Second, we performed an extensive simulation study with *160* parameter combinations mimicking different data-generating processes. Using these synthetic datasets, four additional semi-synthetic datasets and three real datasets, *ConNIS* demonstrated its superiority over five state-of-the-art *Tn5* analysis methods, especially in settings with

low- and medium-dense libraries. In this context, we propose to use a weighting factor when applying genome-wide insertion density values to better represent genomic regions with low insertion densities. This modification also improved three competing methods by reducing the number of false positives without losing too many true positives in many settings. Third, we provide for the first time a data-driven instability criterion for selecting thresholds and parameter values in *TIS* methods, thereby making the results from different studies and methods comparable and more transparent. Applications of this approach to real and synthetic datasets clearly demonstrate its suitability for setting parameter values for all methods considered. An in-depth analysis of biological functions for selected genes illustrates the ability of *ConNIS* to reliably detect essential genes even among short genes that are often excluded from statistical analyses because competing methods cannot distinguish signal from noise in this length range. *ConNIS* and the instability criterion for all competing methods have been made available as an R package. To further explore our results, we provide an interactive web app and publicly available code.

## Materials and methods

### Consecutive Non-Insertion sites (ConNIS)

The transcription of an essential gene is, by definition, vital for an organism's survival and its hindrance due to transposon insertion will result in the mutant's removal from the population. Consequently, IS are expected to be exclusively detected in the non-essential genome, which comprises genes, intergenic regions and smaller fractions of essential genes [7]. Based on the assumption that a *Tn5* transposon can occur at any position in the *non-essential* genome, we propose *ConNIS*, a method that classifies a gene as 'essential' or 'non-essential' by analyzing its largest insertion-free sequence in terms of base pairs. Therefore, we derive a novel probability distribution (see S1 File) that we used to determine the probability of observing an insertion-free sequence in a gene of given length and number of IS.

Let $b$ denote the length of a genome in terms of base pairs that contains $p$ genes with corresponding lengths $b_1, b_2, \ldots, b_p$. We define $\theta = h/b$ as the genome-wide insertion density, where $h$ is the genome-wide number of observed IS. For a gene $j = 1, \ldots, p$ let $\hat{h}_j = \lfloor b_j \cdot \theta \rceil$ be the rounded expected number of IS within gene $j$ under the assumption that gene $j$ is not essential. Furthermore, let $L_j$ be the length of an observed consecutive sequence of non-insertions of gene $j$ under the assumption of uniformly distributed IS (Fig 1A).

Based on Theorem 1 (see Section 1 in S1 File), the probability mass function of $L_j$ is given by

$$\mathbb{P}(L_j = i) = f(i, b_j, h_j) = \frac{\binom{b_j - i - 1}{b_j - \hat{h}_j - i}}{\binom{b_j - 1}{b_j - \hat{h}_j - 1}}.$$

We next consider that IS are often distributed non-uniformly across the genome (see Fig B in S2 File). Other approaches use the genome-wide insertion density $\theta$ to estimate expected insertions per gene. However, under the assumption of non-essentiality this can inflate false positives in regions with a lower-than-average IS density, where missing insertions are likely due to chance. *ConNIS* corrects for this by introducing a weight factor $w$ ($0 < w \leq 1$) to $\theta$ that adjusts for low-density regions (Fig 1B), a bias not handled by normalization methods focused only on insertion counts.

Observing the longest gap of size $l_j$ ConNIS is then defined as the probability of observing an insertion-free consecutive sequence of at least length $l_j$ in gene $j$:

$$\mathcal{C}\left(l_j, b_j, \hat{h}_j, w\right) := \mathbb{P}(L_j \geq l_j) = 1 - \mathbb{P}(L_j < l_j) = 1 - \sum_{i=1}^{l_j - 1} \frac{\binom{b_j - i - 1}{b_j - \lfloor \hat{h}_j w \rceil - i}}{\binom{b_j - 1}{b_j - \lfloor \hat{h}_j w \rceil - 1}}.$$

(1)

**A**

Determination of longest insertion-free sequence for gene *j*

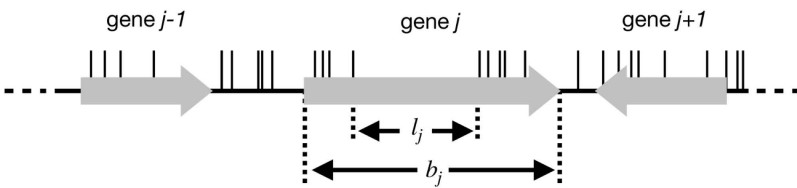

**B**

Determination of the insertion density *θ* and weight value *w*

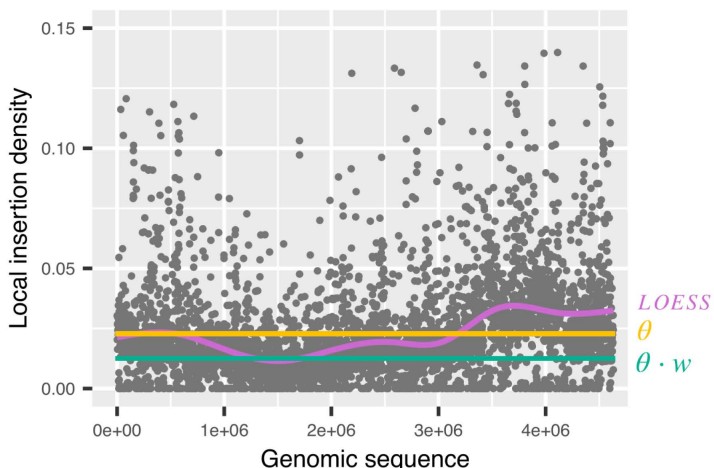

**C**

ConNIS with estimated number of insertion sites and weight

$$
C\left(l_j, b_j, \hat{h}_j, w\right) := 1 - \sum_{i=1}^{l_j-1} \frac{\binom{b_j - i - 1}{b_j - \hat{h}_j w - i}}{\binom{b_j - 1}{b_j - \hat{h}_j w - 1}}
$$

$$
\text{with } \hat{h}_j = b_j \cdot \theta \quad \text{and} \quad 0 < w \le 1
$$

**Fig 1. Overview of *ConNIS*. A** For a gene *j*, determine its length $b_j$ and its longest insertion-free sequence $l_j$. **B** Set a weight *w* for the genome-wide insertion density *θ* reflecting rather low density regions of the genome. **C** *ConNIS*: The probability of observing $l_j$ within gene *j* due to random chance.

Given a significance level of $0 < \alpha < 1$, we declare a gene *j* to be essential if $C\left(l_j, b_j, \hat{h}_j, w\right) \le \alpha$. To control the global type I error when applying *ConNIS*, we suggest using either the Bonferroni(-Holm) method [39,40] to control the family-wise error rate (FWER) or the Benjamini-Hochberg method [41] to control the false discovery rate (FDR).

**Competing state-of-the-art methods with proposed weighting strategy.** We compared *ConNIS* with five popular state-of-the-art *Tn5* analysis methods for determining essential genes:

1. the *Binomial* distribution approach in the TSAS 2.0 package [30],

2. the approach of fitting a bimodal distribution based on gene-wise insertion densities included in the Bio-TraDIS package [32] (referred to as *Exp. vs. Gamma* method throughout this paper),

3. the *InsDens* method [33],

4. the *Tn5Gaps* method of the TRANSIT package [31] and

5. the *Geometric* distribution.

Although the geometric distribution has not been published as a stand-alone method, we included it as a competitor because it is the limiting distribution of *ConNIS* (see Theorem 2 in S1 File) and has been part of an analysis pipeline for determining the probability of insertion-free regions in the genome [42].

The *Binomial* and *Geometric* methods use the genome-wide insertion density $\theta$ as success probability, and the *Tn5Gaps* method uses it as a location parameter. However, for the reason outlined above, this naive use of $\theta$ can increase the number of false positives within genomic regions with a relatively low insertion density compared to the rest of the genome. To address this potential pitfall, we introduce a weight $w$ to adjust $\theta$ when applying these methods, as we do it in *ConNIS*. We then use these modified methods, as well as the original versions, for comparison. See Sect 2 in S1 File for methodological details. Further, *InsDens* requires several prior hyperparameters. In line with the authors' claim, our tests on selected simulation settings showed minimal impact from these choices [33]. Thus, we used the default settings of the R package insdens for all simulations and real data analyses (see https://github.com/Kevin-walters/insdens, commit 286f114).

**A labeling instability criterion for tuning parameter selection.** TIS methods often require *a priori* set parameter or threshold values that will influence the final number of genes labeled 'essential' and therefore the methods' performances in terms of correct classification. The setting of an 'appropriate' parameter/threshold value in a given data scenario can be interpreted as a *tuning* problem. In this context our data-driven tuning approach selects a parameter of threshold value for a TIS method from a set of candidate values.

We consider observed IS as realizations of an unknown probability distribution across the genome. This is comparable with a repeated TIS experiment which yields different IS positions in each realization, particularly in non-essential genomic regions. As a result, gene-wise IS metrics, such as the longest insertion-free sequence $l_j$ or gene-wise IS density $\theta_j$, would vary between experiments, potentially altering the set of genes classified as essential. The main idea of our selection criterion is to leverage these variations by quantifying the average variation of gene labeling based on $m$ subsample for a given tuning value. Inspired by stability selection approaches in linear regression and graph estimation problems [43–45], a 'good' tuning value should give rather stable results in terms of gene classification, i.e., the gene labeling should be less sensitive to the random occurrence of IS across the genome. In the following, we detail the procedure for selecting a suitable weight value $w$ using *ConNIS* as an example. A transfer to other *TIS* methods or to threshold based filters in the data pre-processing steps is straight forward.

Let $\mathbf{w} = w_1, w_2, \ldots, w_z$ be a sequence of ordered weights ($0 < w_q < w_r \leq 1$ for $q < r$ and $w_q, w_r \in \mathbf{w}$). Assume further that $m$ subsamples, each of of size $h^{sub} < h$, are drawn without replacement from the set of $h$ observed IS and the expected number of insertion sites per gene to be $\hat{h}_j^{sub} = b_j \frac{h^{sub}}{b}$ (Fig 2A).

For all $w_y \in \mathbf{w}$, genes are labeled as 'essential' or 'non-essential' for a given significance level within each of the $m$ subsamples using *ConNIS*. By modeling the gene labeling as a Bernoulli process ('essential' or 'non-essential'), we estimate the probability of labeling a gene $j$ as 'essential' (see Fig 2B) using

## A

### Drawing of $h^{sub}$ insertion site subsamples

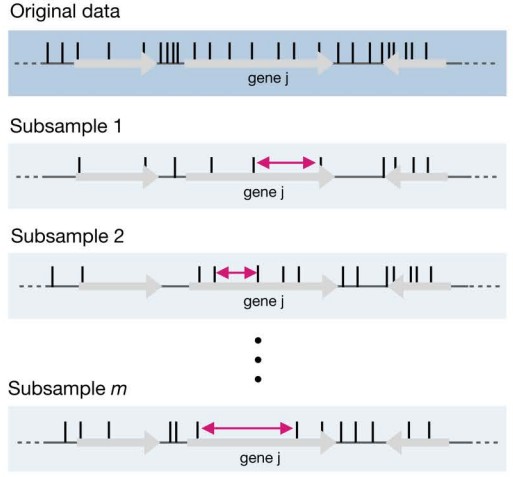

## B

### Instability values based on subsamples

$$\hat{\pi}_j(w_y) = \frac{1}{m} \sum_{d=1}^{m} \mathbf{1}\left[ C\left( l_j^{(d)}, b_j, \hat{h}_j^{sub}, w_y \right) \leq \alpha \right]$$

$$\phi(w_y) := \sum_{j=1}^{p} \frac{\hat{\pi}_j(w_y) \cdot \left( 1 - \hat{\pi}_j(w_y) \right)}{q(w_y)}$$

for $w_y \in \mathbf{w}$ with $\mathbf{w} = \{w_1, w_2, \ldots, w_z\}$

## C

### Determination of $w_s \in \mathbf{w}$

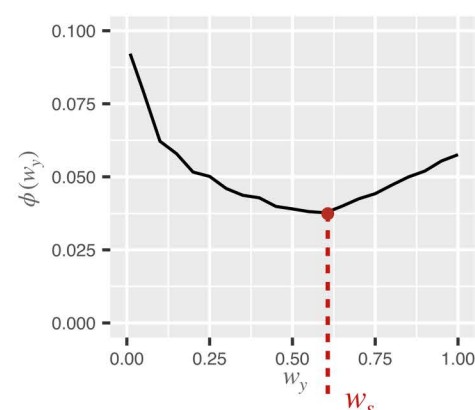

## D

### Application of ConNIS with selected weight

$$C\left( l_j, b_j, \hat{h}_j, w_s \right) \quad \text{for } j = 1, 2, \ldots, p$$

| gene | probability gap l | adjusted $\alpha$ | putative essential |
|------|-------------------|-------------------|--------------------|
| .. | … | … | … |
| j-1 | 0.0045 | 0.00005 | FALSE |
| j | 0.000007 | 0.00005 | TRUE |
| … | … | | … |

**Fig 2. Overview of the labeling instability criterion to select the weight parameter for *ConNIS*. A** Drawing $m$ subsamples of the $h$ original observed IS. **B** Calculation of the instability values for all weights $w_y \in \mathbf{w}$ based on the estimated variances of a Bernoulli variable. **C** Selecting the weight $w_s$ with the lowest instability $\phi(w_y)$. **D** Application of *ConNIS* using $w_s$ followed by a multiple testing correction to identify putative essential genes.

$$\hat{\pi}_j(w_y) = \frac{1}{m} \sum_{d=1}^{m} \mathbb{1} \left[ \mathcal{C} \left( l_j, b_j, \hat{h}_j^{sub}, w_y \right) \le \alpha \right].$$

We can then define the *instability criterion* over all genes for a given weight $w_y$ as

$$\phi(w_y) := \sum_{j=1}^{p} \frac{\hat{\pi}_j(w_y) \cdot \left( 1 - \hat{\pi}_j(w_y) \right)}{q(w_y)}, \tag{2}$$

where $\hat{\pi}_j(w_y) \cdot \left( 1 - \hat{\pi}_j(w_y) \right)$ is the Bernoulli variance and $q(w_y) = \sum_{j=1}^{p} \mathbb{1} \left( \hat{\pi}_j(w_j) > 0 \right)$ is the total number of genes that have been labeled at least once as 'essential' in the $m$ subsamples. This normalization factor ensures $0 \le \phi_j(w_y) \le 0.25$. A value of $\phi(w_y) = 0$ indicates complete consistency in gene labeling for each gene across subsamples, whereas $\phi(w_y, d) = 0.25$ reflects total instability, equivalent to randomly assigning labels by flipping a fair coin for each gene in each subsample.

After calculating the instability for all weights, we have a sequence of instability values $\phi = \{\phi(w_y)\}_{y=1}^{z}$ and select then the weight $w_y$ that minimizes the instability of labeling (see Fig 2C):

$$w_s = \underset{w_y \in \mathbf{w}}{\arg\min} \, \phi(w_y).$$

Finally, *ConNIS* is applied to the original data with $w_s$ (Fig 2D).

Depending on the range of $\mathbf{w}$ and the number of observed IS, it is possible that very small weight values may lead to instability values approaching or even reaching zero with (nearly) all genes being labeled as 'non-essential'. Following other stability approaches [43–45], these values are excluded from the set of candidate tuning values because they provide no useful information. For our instability criterion, we propose to omit all weights smaller than the smallest weight that maximizes the function $\phi(w)$ to ensure that only the most informative weights are considered, i.e.,

$$w_{max} = \min \left\{ w_y \in \mathbf{w} \mid \underset{w_y \in \mathbf{w}}{\arg\max} \, \phi(w_y) \right\}$$

$$w_s = \underset{\tilde{w}_y \in \mathbf{w}: \tilde{w}_y \ge w_{max}}{\arg\min} \, \phi(\tilde{w}_y).$$

## Results

### Comparison of ConNIS with state of the art methods

To evaluate the performance of *ConNIS* and its competitors we applied all methods to synthetic, semi-synthetic and real-world data. A detailed explanation of our simulation schemes for generating synthetic data that covers different assumptions about the data generating process can be found in S3 File. For the application to real-world data we used three publicly available *Tn5* libraries of different organisms and insertion densities. The semi-synthetic datasets were generated from a high-density *Tn5* library by randomly deleting IS. Note that the performance of all methods depends on the chosen values of their respective parameters and thresholds.

Since the number of essential genes is small compared to the number of non-essential genes, we used the Mathew's Correlation Coefficient (MCC), a metric suitable for imbalanced data [46–48], as main performance measure. The MCC equals 1 when the method perfectly labels all genes as 'essential' or 'non-essential' (perfect agreement), 0 when the

labeling is completely random, and –1 when there is perfect disagreement between the true and predicted labels. For comparison the MCC is plotted given the number of genes labeled 'essential' which can be controlled by the methods' different parameter and threshold values. In addition, the precision-recall-curve (PRC) is shown to investigate two desirable, yet occasionally, conflicting objectives: selecting as many true positives as possible (recall) while avoiding an inflated number of false positives (precision).

In all applications the weight value of *Binomial*, *ConNIS*, *Geometric* and *Tn5Gaps* was set to $w = 0.1, 0.2, \ldots, 1$ with $w = 1$ being the original, unweighted version. For *Exp. vs. Gamma*, we set the $\log_2$-likelihood ratio threshold $t \in \{2, 3, \ldots, 12\}$, covering the range of values commonly reported in the literature. The posterior probability threshold of *InsDens* was set at $r \in \{0.01, 0.1, 0.2, \ldots, 0.9, 0.99\}$. Furthermore, we truncated genes by excluding the distal ends by either 0%, 5% or 10% and applied Bonferroni(-Holm) and Benjamini-Hochberg procedures for multiple testing correction.

**Synthetic data settings.** We present three illustrative synthetic data settings offering an overview of performance the methods. The results of all 160 different settings and additional classification metrics can be interactively explored at https://connis.bips.eu, supporting our findings.

**Synthetic data example 1 (SDE1)** had 200,000 IS randomly distributed along the genome in a sinusoidal shape. 'Essential' genes were defined to contain insertion-free sequences of at least 75% of the gene's length. This represented scenarios where essential genes can contain IS relatively far from their distal ends. Fig 3A shows *ConNIS* clearly outperforming the other methods with regard to the MCC. The PRC demonstrates that *ConNIS* effectively enhanced the identification of true essential genes without substantially inflating false positives by maintaining high precision. Notably, all methods showed their peak performance when the number of genes labeled 'essential' was about the number of true essential genes (dashed orange line). The plot also highlights the beneficial effect of applying a weight $w < 1$ to *Binomial* and *Geometric* (the points indicate the average performance if no weight is applied, i.e., $w = 1$).

**Synthetic data example 2 (SDE2)** mimicked scenarios where only a low insertion density can be achieved, e.g., due to bottleneck effects by environmental pressure. Therefore, 50,000 IS were randomly distributed along the genome in a sinusoidal shape, and the essential genes had insertion-free sequences of at least 85% of their length. All methods suffered from the sparse library (Fig 3C). *ConNIS* achieved the highest MCC value if the number of genes labeled 'essential' was close to the number of true essential genes. *Binomial* and *Tn5Gaps* could only achieve mediocre values at best. For *Exp. vs. Gamma*, *InsDens* and *Tn5Gaps*, the range of the number of genes labeled 'essential' never contained the number of true genes. However, the first two could achieve MCC values that were slightly worse than *ConNIS*. With respect to the PRC, all methods induced false positives due to larger non-insertion sequences occurring by chance compared to denser libraries. *ConNIS* tended to have a rather high precision while *Exp. vs. Gamma* and *InsDens* achieved rather high recall values, but at the price of an inflation of false positives.

**Synthetic data example 3 (SDE3)** covered scenarios with so-called 'cold-spots' along the genome, which have a much lower chance of containing IS. 'Essential' genes were defined to contain insertion-free sequences of at least 80%. In non-essential sections of the genome, each base pair had the same probability to contain one of the 200,000 IS, yet, in *25* randomly placed sections of size 10,000bp these probabilities were lowered by factor *10*. *ConNIS* achieved clearly the best MCC values and PRC performance (Fig 3D). However, all methods tended to overestimate the number of essential genes (indicated by the rather low precision values) due to the higher chance of false positives in cold spots.

**Real-world data.** In the first example, we applied all methods to an *E. coli* BW25113 strain library comprising approximately 102,000 IS at time point T0 [49]. As ground truth, we used the results of the single-knockout study by Baba et al. [4], which is often considered as the gold standard. Fig 4A shows *ConNIS* outperforming the other methods by reaching MCC values up to 0.65. *InsDens* and *Exp. vs. Gamma* labeled too many genes as 'essential' (at least 575) even for their strictest thresholds ($t = 12$ and $r = 0.99$). However, the thresholds of *Exp. vs. Gamma* had only a marginal influence on the number of genes labeled 'essential', which resembles in parts the results of the simulation study. The

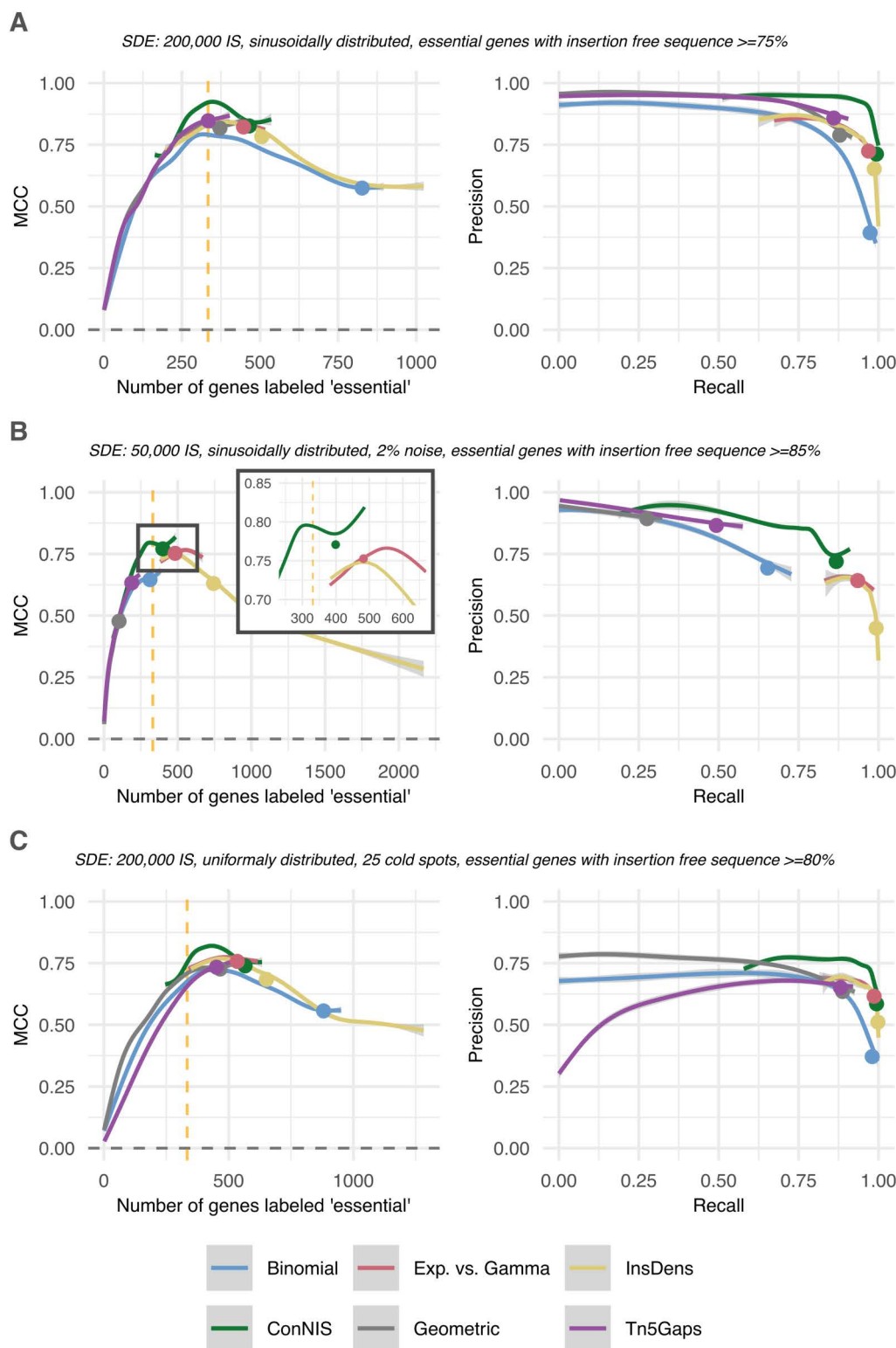

**Fig 3. MCC and PRC performance based on synthetic data.** The plots show the LOESS-smoothed curves with 95% confidence intervals for the MCC and PRC in three synthetic data settings. At the vertical dotted line the number genes labeled 'essential' matches the number of true essential genes. Dots on the curves indicate the average performance without weights or the least stringent threshold ($t = 2$ for *Exp. vs. Gamma* and $r = 0.1$ for *InsDens*). In all settings, 5 % of both ends of each gene were trimmed.

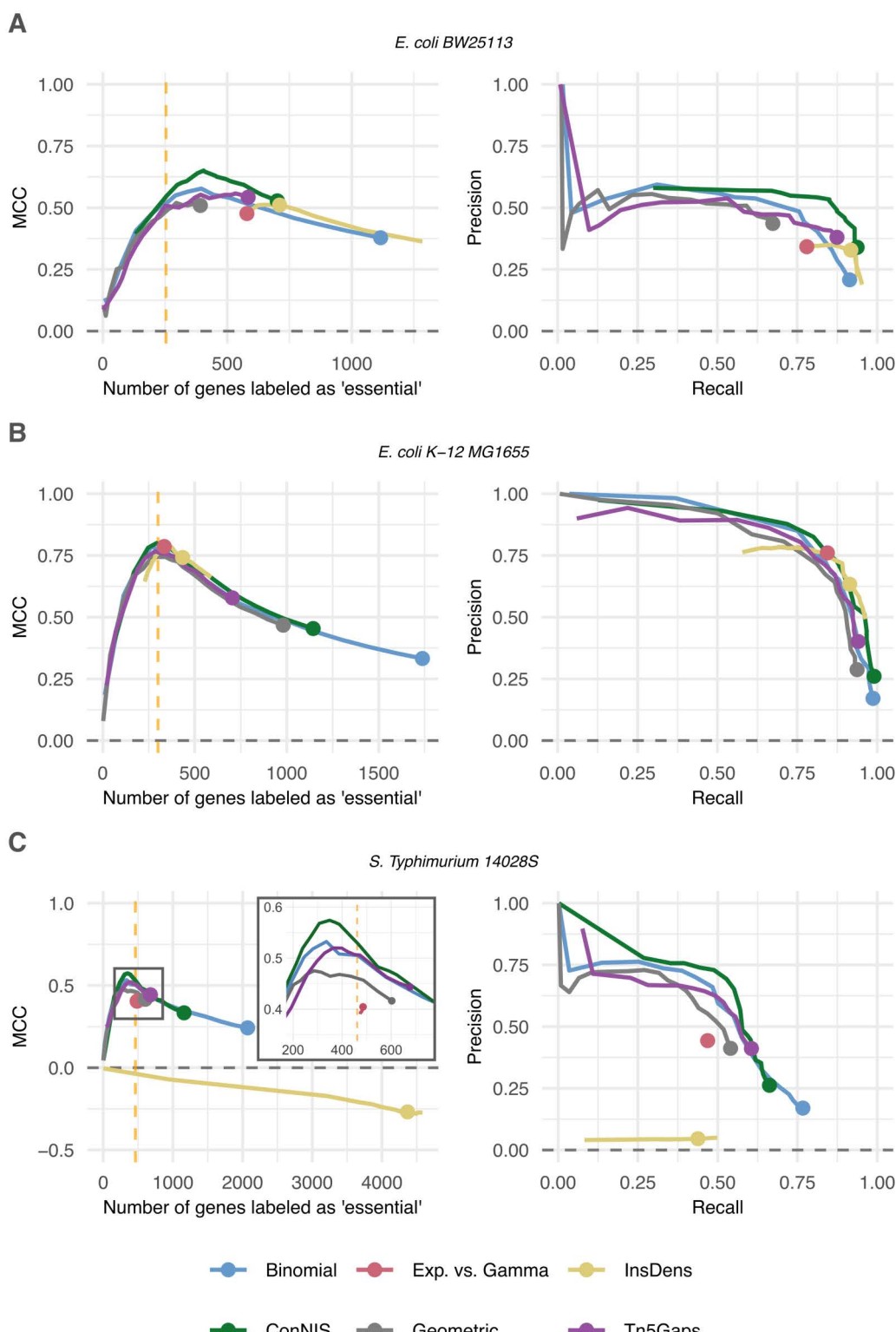

**Fig 4. MCC and PRC performances based on real-world data.** The vertical dotted line shows the true number of genes. Dots indicate the performance of the original methods (*w* = 1). **A** *E. coli* BW25113 strain with ≈102,000 IS [49]. **B** *E. coli* K-12 MG1655 strain with ≈390,000 IS [50]. **C** *Salmonella enterica* serovar Typhimurium 14028S strain with ≈186,000 IS [37]. Note, in applications A and B, most of the results of *Exp. vs. Gamma* are covered by those of *InsDens*.

PRCs reveal that all methods achieved mediocre precision values at best with *ConNIS* having relatively stable precision values for rising recall values.

A high-density *E. coli* K-12 MG1655 library characterized by approximately 390,000 IS [50] was used as seccond example. As truth, we used the gene essentiality classification from the Profiling of *E. coli* Chromosome (PEC) database [51]. Here, results were consistent with the simulation study, i.e., all methods benefited from the high number of IS and performed best when the number of selected genes approached the number of true essential genes (Fig 4B). Weighting ($w < 1$) was highly beneficial for all methods, as the original versions ($w = 1$, indicated by the points) had high recall values at the cost of low precision values.

In the third example, all methods were applied to a *Salmonella enterica* serovar Typhimurium 14028S library comprising approximately 186,000 IS [37]. Following Nlebedim et al. [33], we used as truth the combined set of essential genes provided by Baba et al. [4] and Porwollik et al. [52]. In this scenario all methods showed at best mediocre performance. The removal of IS with low read counts improved the performances of the methods slightly and might be a sign of the presence of spurious IS [31,53] (we tried minimum read count thresholds with value *1*, *2*, *3*, *5* and *10*). *ConNIS* achieved the best MCC and precision values. *Exp. vs. Gamma* and especially *InsDens* performed very poorly (Fig 4C), with the latter labeling far too many genes as 'essential', resulting in negative MCC values. *Binomial*, *ConNIS*, and *Tn5Gaps* benefited from a fairly low weight $w$, which seemed to reduce the number of false positives comapred to the original versions ($w = 1$).

**Semi-synthetic data settings.** To investigate the influence of the number of observed IS on the methods' performances, we generated semi-synthic data by drawing IS subsamples of sizes $50,0000$, $100,000$, $200,000$ and $400,000$ from a very high density *Tn5* library [11]. The Kaio library [4] was used as a reference for true gene essentiality. In low and medium density libraries (subsamples of $50,000$ to $200,000$ IS) *ConNIS* outperformed the other methods, clearly (see Fig 5A; for medium-sized libraries see Fig A in S2 File). Similar to the synthetic and real-world data settings, *ConNIS* showed its best performance in terms of MCC when the number of selected genes was about the number of true essential genes. In case of the rather high-density library (subsample size of $400,000$ IS), all methods were on par, (Fig 5B).

### Application of gene labeling instability criterion for tuning parameter selection

The performance of the gene labeling instability criterion for tuning parameter selection was investigated using the three previously described real-world datasets and three randomly chosen dataset examples from the simulation study. We applied the instability criterion to select the tuning parameter of each method. For each setting, $m = 500$ subsamples were drawn without replacement, with each subsample containing 50% of randomly picked IS from the original data. Genes were truncated by $5\%$ at each distal end. We used the MCC to evaluate the performance of the labeling instability criterion and compared it to MCC values for the *optimal* parameters (those that produce the highest MCC) as well as for parameters used in earlier studies, such as unweighted versions or heuristic choices.

The results in Table 1 demonstrate that the application of the gene labeling instability criterion for determining a weight for *ConNIS* is highly beneficial. For the real-world datasets *E. coli* BW25113 and *Salmonella enterica* serovar Typhimurium 14028S, as well as the synthetic dataset 2, the application of instability criterion successfully identified the 'optimal' weights based on the corresponding MCC values (Table 1). In these examples, *ConNIS* also achieved the highest MCC with its selected $w$ compared to all other methods. Furthermore, for synthetic datasets 1 and 3, the MCCs obtained by the instability criterion were close to its highest possible MCCs in these settings (0.90 vs. 0.94 and 0.76 vs. 0.77). Only for the MG1655 strain data, our tuning approach was less successful for *ConNIS* (0.67 vs. 0.79).

The instability criterion also successfully tuned the other methods, resulting in many cases where the best possible MCC was achieved. For *Exp. vs. Gamma* in all six settings $\log_2$ thresholds were selected that resulted in MCC values close or even to values obtained by applying an optimal threshold value. In comparison to $\log_2$ thresholds used in recent studies

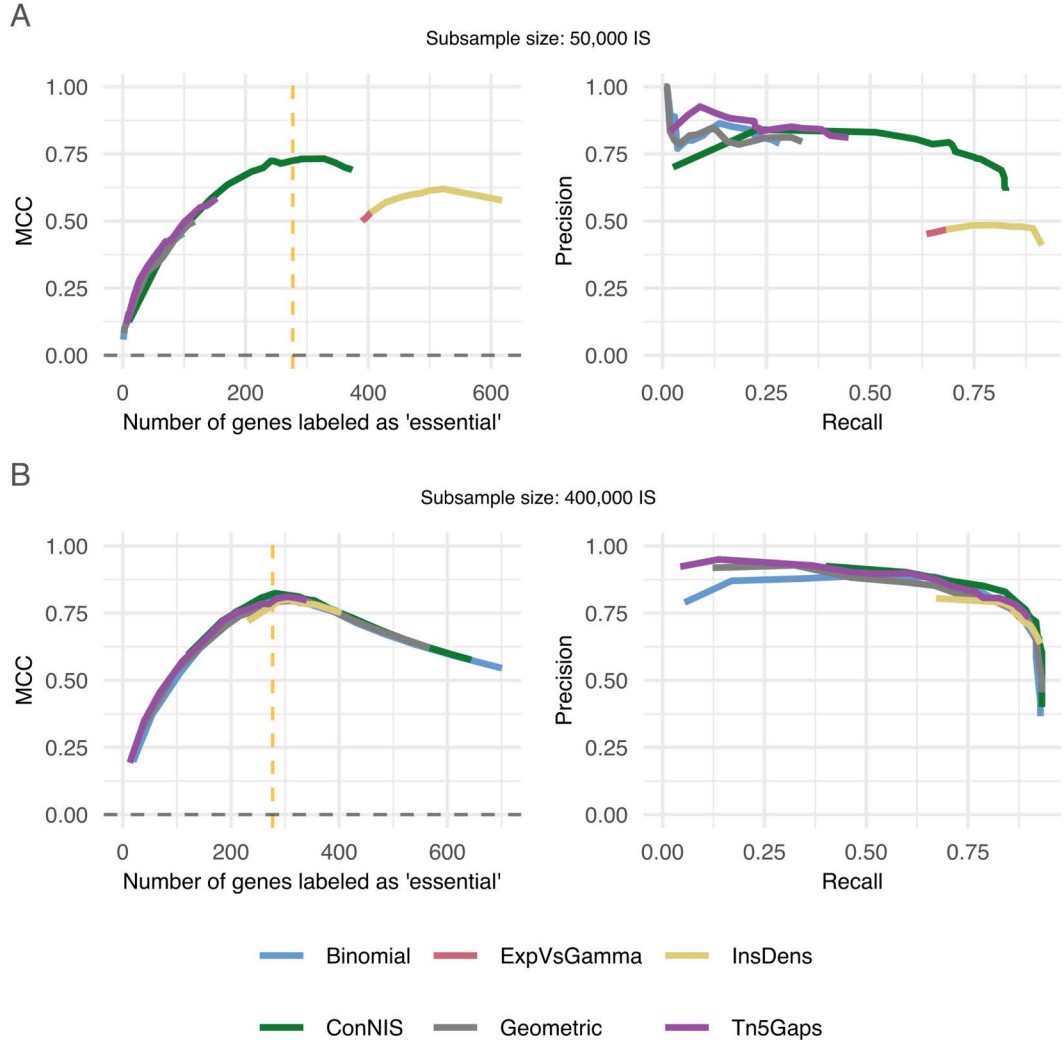

**Fig 5. MCC and PRC performances for semi-synthetic data.** Subsamples were generated by randomly drawing IS from a very high-density library to generate a low- (**A**) and high-density (**B**) library of *E. coli* BW25113 [11]. The Kaio library [4] was used as reference for 'true' gene essentiality. The vertical dotted line indicates where the number of genes labeled "essential" corresponded with number of true essential genes.

[11,20] we found our instability approach to give similar or better MCC values, yet, the range of possible MCC was rather small. Applying labeling instability to the posterior probability threshold *r* in *InsDens* yielded favorable results in five settings and achieved the highest possible MCC in three of them. It was also able to select a good choice between a very strict (*r* = 0.99) and a very relaxed value (*r* = 0.05) which have been used before [33]. Only in the *Salmonella enterica* serovar Typhimurium 14028S real-world dataset, the selected *r* had a bad MCC value. However, since *InsDens* generally showed a weak performance in this setting before (Fig 4C), the result is not surprising. For *Binomial*, *Geometric* and *Tn5Gaps* the application of the instability approach was also beneficial compared to the unweighted (i.e., original) version of the methods.

## Biological relevance

To highlight biological relevance beyond global performance metrics, we analyzed genes where *ConNIS* systematically disagrees with the other methods. Here, we focus on 'major discrepancies', i.e., genes called 'essential' by ConNIS but

**Table 1. Tuning performance of the gene labeling instability criterion.**

| | | BW25113 | MG1655 | 14028S | Syn. Data 1 | Syn. Data 3 | Syn. Data 3 |
|---|---|---|---|---|---|---|---|
| **Binomial** | instability | 0.55 | 0.71 | 0.51 | 0.83 | **0.73** | 0.61 |
| | optimal | 0.58 | 0.79 | 0.53 | 0.89 | 0.73 | 0.69 |
| | unweighted | 0.38 | 0.33 | 0.24 | 0.51 | 0.64 | 0.52 |
| **ConNIS** | instability | **0.64** | 0.67 | **0.57** | 0.9 | **0.88** | 0.76 |
| | optimal | 0.64 | 0.79 | 0.57 | 0.94 | 0.88 | 0.77 |
| | unweighted | 0.17 | 0.08 | 0.33 | 0.81 | 0.34 | 0.21 |
| **Exp. vs. Gamma** | instability | 0.47 | 0.78 | 0.39 | 0.91 | **0.81** | 0.72 |
| | optimal | 0.49 | 0.79 | 0.41 | 0.92 | 0.81 | 0.74 |
| | $\log_2(4)$ | 0.47 | 0.77 | 0.39 | 0.88 | 0.81 | 0.73 |
| | $\log_2(12)$ | 0.47 | 0.71 | 0.39 | 0.91 | 0.74 | 0.73 |
| **Geometric** | instability | **0.52** | 0.58 | 0.46 | 0.86 | 0.59 | 0.63 |
| | optimal | 0.52 | 0.74 | 0.48 | 0.91 | 0.75 | 0.70 |
| | unweighted | 0.22 | 0.08 | 0.42 | 0.83 | 0.42 | 0.22 |
| **InsDens** | instability | **0.51** | 0.74 | −0.26 | 0.91 | **0.81** | **0.74** |
| | optimal | 0.51 | 0.78 | 0.00 | 0.92 | 0.81 | 0.74 |
| | $r = 0.05$ | 0.48 | 0.72 | −0.27 | 0.81 | 0.74 | 0.63 |
| | $r = 0.3$ | 0.51 | 0.78 | −0.25 | 0.86 | 0.79 | 0.70 |
| | $r = 0.6$ | 0.49 | 0.76 | −0.22 | 0.90 | 0.81 | 0.73 |
| | $r = 0.99$ | 0.47 | 0.64 | − | 0.90 | 0.74 | 0.72 |
| **Tn5Gaps** | instability | 0.55 | 0.63 | **0.52** | **0.91** | **0.77** | **0.70** |
| | optimal | 0.56 | 0.77 | 0.52 | 0.91 | 0.77 | 0.70 |
| | unweighted | 0.54 | 0.58 | 0.44 | 0.91 | 0.72 | 0.60 |

Tuning was applied to three real-world and three synthetic datasets. For each method, we report the MCC obtained when using the gene labeling *insta-bility* criterion. For comparison, we also show the MCC achieved using (i) the *optimal* tuning value (i.e., the "oracl" value yielding the highest possible MCC) and (ii) heuristic values used in previous studies. The entries in bold indicate cases where the MCC based on the instability criterion was identical to the MCC value of the optimal tuning value. **Syn. Data 1**: 400,000 sinusoidal distributed IS, essential genes contained an insertion free sequence of $\geq$ 80%. *8000* IS where added as noise. **Syn. Data 2**: 100,000 sinusoidal distributed IS and essential genes contained an insertion-free sequence of $\geq$ 75%. **Syn. Data 3**: 200,000 uniformly distributed IS with 25 cold spots, *4000* noise IS and essential genes contained an insertion-free sequence of $\geq$ 75%.

"non-essential" by four to five comparator methods, or vice versa. Across the three libraries (*E. coli* BW25113, *E. coli* MG1655 and *S. Typhimurium* 14028s), this affects 59 genes in total (15, 26 and 18 genes, respectively). Of these, 44 genes are called 'essential' by *ConNIS* but 'non-essential' by the comparator methods, whereas 15 genes show the oppo-site case (see S4-S6 Files). Overall, the analysis shows that *ConNIS* agrees well with experimental gold-standard essenti-ality sets while providing specific gains in low-insertion regimes and for short genes that have often been excluded *a priori* from the analysis due to lack of detection power of established methods. For all methods the threshold/parameter values were set by our instability criterion.

In the first group, *ConNIS*-specific essential calls have a median length of 328 bp (interquartile range *131* to *477*bp; minimum gene length *74*bp). For example, *ftsL* (*365*bp), ffs (*113* bp), *argU* (*76*bp), and *folK* (*479*bp) were correctly iden-tified as being essential by *ConNIS*. *FtsL* encodes a cytoplasmic membrane protein which essentiality manifests in rapid cell division blockade upon mutation [54]. *Ffs* together with *Ffh* builds up the well-known signal recognition particle (SRP) in *E. coli*, a multifunctional ribonucleoprotein complex fundamental for membrane protein targeting. As described by

Peterson et al. [55], both the *Ffh* protein and the *ffs* encoded 4.5S RNA are essential for cell viability and correct localization of proteins to the cytoplasmic membrane. Another RNA-gene correctly identified by *ConNIS* is *argU*. Lack of function mutations have been reported to cause DNA replication defects [56] manifesting in inhibition of cell growth [57]. Essentiality of *folK* has been critically analyzed by Goodall et al. [11] who, in contrast to the Keio library [4] and PEC database [51], classified the gene as 'conditionally essential' and not as 'essential'. However, *ConNIS* also named *folK* essential. Interestingly, Goodall et al. [11] as well as the Keio library [4], the PEC database [51] and *ConNIS* are correct and the explanation highlights the importance of the chosen growth conditions on which basis essentiality is defined. While Goodall et al. [11] determine essentiality by using a library obtained directly from LB-agar plates and conditional essentiality by a library obtained after successive growth to 5–6 generations in liquid LB medium, Wetmore et al. [49], which data were used in this study, grew their mutant library on LB-plates followed by growth in liquid LB medium (as Goodall et al. [11] to an OD of 1.0). Consequently, *ConNIS* identifies *folK* correctly as being essential for the Wetmore et al. [49] mutant library. The disagreement pattern between *ConNIS* and its competitors is consistent with the known limitation of density/count-based methods on short or low-insertion genes, which are often excluded or down-weighted. Thus, the results highlights that *ConNIS* retains statistical power even for short genes. Yet, in the case of *nusB*, one of the four Nus factor encoding genes of *E. coli*, *ConNIS* wrongly assigns 'essentiality', while Bubunenko et al. [58] have shown that *NusB*, despite of being important for cell growth, is not essential. A closer inspection revealed that the incorrect assignment resides in the fact that the analyzed library carries only one insertion at the far 3'-end of *nusB*. Consequently, the relatively long insertion-free gap yields a relatively low *p*-value.

The second group comprises genes that *ConNIS* called 'non-essential' but that $\geq$ 4 the other methods classified as 'essential'. These genes are typically much longer (median length $1,440$bp, interquartile range *1132* to *1659*bp). Two examples of genes correctly identified as being non-essential by *ConNIS* are *ptsI* and *ybcK*. As shown by Wu et al. [59] and Wu et al. [60] viable loss-of-function mutants of *ptsI* and *ybcK*, respectively, can be recovered. On the other hand, *ConNIS* classified *pssA* as non-essential, whereas the encoded phosphatidylserine synthase (*PssA*) is known to be essential for vitality in various pathogenic bacteria including *E. coli* [61]. The wrong assignment can be explained by the combination of three factors: first, only one insertion was observed close to the middle of the gene making the observed gap nearly as small as possible for a single insertion site. Second, a rather low number of expected insertions sites was used due to the low weighting factor of $w = 0.15$, increasing the probability to observe bigger insertion free gaps under the null model. Third, the applied Bonferroni-Holm correction method is relatively conservative and can label even small *p*-values non-significant when thousands of genes are examined.

## Discussion and conclusion

In this work, we addressed three main challenges inherent in statistical analysis in *TraDIS* studies. The first challenge arises from the fact that in *Tn5* datasets every base pair of the genome serves a potential insertion site, while reported insertion densities often remain far below saturation levels. Considering this, *ConNIS* gives an analytic solution for the probability of observing an insertion-free sequence within a gene of a given length and number of insertion sites. The second challenge is the often observed non-uniform distribution of IS across the genome. Neglecting this factor can lead to an increased number of (nearly) insertion-free genes being incorrectly labeled as essential in regions with relatively low insertion densities. Addressing non-uniformity, *ConNIS* contains a weighting parameter that increases the precision by making it more difficult to label genes as 'essential' in low-density regions. We extended this idea to three state-of-the-art methods to improve their precision. The third challenge lies in the fact that many *TIS* methods rely on *a priori* set threshold or parameter values, which can substantially influence labeling performance. However, an 'objective' criterion for setting these values has been lacking, often resulting in arbitrarily chosen values. By introducing the concept of gene labeling instability based on subsamples of observed IS, we proposed a data-driven approach to select appropriate parameters and threshold values.

An extensive simulation study and application to three real-world datasets and four semi-synthetic datasets was conducted to compare the performance of *ConNIS* to multiple state-of-the-art *Tn5* analysis methods. In most settings, *ConNIS* outperformed these methods or was at least on par with the best of them. Unlike its competitors, *ConNIS* showed usually robust performances for (arbitrarily) chosen truncation and filter values. The results also confirm our idea of weighting the genome-wide insertion density when applied to existing methods: it could reduce the number of false positives without sacrificing too many true positives. Applying our proposed gene labeling instability criterion for tuning parameter selection in various real-world and multiple synthetic data scenarios demonstrated its potential to select favorable weight and threshold values for all methods. By inspecting major discrepancies between the classification results of *ConNIS* and its competitors, we showed that *ConNIS* was able to correctly classify even very short genes, thereby avoiding the standard practice of dismissing such genes from the analysis *a priori*.

Given that *ConNIS* demonstrated superior performance, especially in low and medium insertion density settings, its application is expected to improve the precision of results in experimental settings characterized by high selective pressure or observation of bottleneck effects. While we have investigated *ConNIS*' ability to identify essential genes, we anticipate that its application might be similarly beneficial for the determination of *conditionally* essential genes, for example by comparing gene-wise *ConNIS* scores between conditions and by defining quasi-essential genes based on differences in these scores between time points or conditions. This would broaden the scope of *ConNIS* to settings where a non-binary characterization such as relative gene fitness is desirable. As a first step, we provide a proof-of-concept (see S7 File), where we illustrate how *ConNIS* in combination with the instability approach yields a continuous gene-wise essentiality evidence score and how this score can be used to classify quasi-essential genes and fitness-like effects. This framework could be extended in future work by explicitly modeling the loss of insertion sites over time or across conditions [27]. Further, the weighting approach could be improved by incorporating multiple weighting values to target different genomic regions more effectively.

Our gene labeling instability criterion was originally developed for selecting threshold and parameter values of *Tn5* analysis methods, but it may also be applicable to other *TIS* methods that employ alternative transposons, such as the popular *mariner* transposon. It might also serve as a criterion in pre-processing steps like quality filters or trimming of distal gene ends. Last but not least, our work showed the crucial role of the underlying data-generating process on the performance of all methods. Future work could expand the range of scenarios considered, helping researchers choose the most appropriate method for analyzing their data. In this context, a systematic re-analysis of publicly available *TraDIS* datasets could raise the confidence in essential gene prediction and allow for further hypothesis generation. As a first step, we provide a curated multi-study resource comprising eight publicly available *Tn5*-based *TraDIS*/*Tn-Seq* datasets, including transparent per-study processing scripts and the resulting *ConNIS* essential-gene predictions, via Zenodo (DOI: https://doi.org/10.5281/zenodo.18538449).

## Supporting information

**S1 File. Proofs and methodological extensions.** PDF file with proofs for *ConNIS* and formal definition of the extension of existing methods.
(PDF)

**S2 File. Additional plots.** PDF file with plots of additional analysis results.
(PDF)

**S3 File. Detailed description of the generation of synthetic data.**
(PDF)

**S4 File. Gene classifications for E. coli BW25113.**
(CSV)

**S5 File. Gene classifications for E. coli MG1655.**
(CSV)

**S6 File. Gene classifications for S. Typhimurium 14028S.**
(CSV)

**S7 File Proof-of-concept for gene fitness and quasi-essentiality.** An R Cookbook.
(PDF)

## Acknowledgments

The authors would like to thank Ian Henderson, Emily Goodall and Ash Robinson for providing the list of insertion sites of their high-density library of the *E. coli* BW25113 strain.

## Author contributions

**Conceptualization:** Moritz Hanke.

**Data curation:** Moritz Hanke.

**Formal analysis:** Moritz Hanke, Ronja Foraita.

**Investigation:** Moritz Hanke, Theresa Harten.

**Methodology:** Moritz Hanke.

**Software:** Moritz Hanke.

**Validation:** Moritz Hanke, Theresa Harten, Ronja Foraita.

**Visualization:** Moritz Hanke, Theresa Harten.

**Writing – original draft:** Moritz Hanke, Theresa Harten, Ronja Foraita.

**Writing – review & editing:** Moritz Hanke, Theresa Harten, Ronja Foraita.

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
