## [Decision Letter · Decision Letter 0]

2 Nov 2025

ConNIS and labeling instability: new statistical methods for improving the detection of essential genes in TraDIS libraries

PLOS Computational Biology

Dear Dr. Hanke,

Thank you for submitting your manuscript to PLOS Computational Biology. After careful consideration, we feel that it has merit but does not fully meet PLOS Computational Biology's publication criteria as it currently stands. Therefore, we invite you to submit a revised version of the manuscript that addresses the points raised during the review process.

Please submit your revised manuscript within 60 days Jan 02 2026 11:59PM. If you will need more time than this to complete your revisions, please reply to this message or contact the journal office at ploscompbiol@plos.org. Please include the following items when submitting your revised manuscript:

We look forward to receiving your revised manuscript.

Kind regards,

Jinyan Li

Academic Editor

PLOS Computational Biology

Ilya Ioshikhes

Section Editor

PLOS Computational Biology

**Journal Requirements:**

**Reviewers' comments:**

Reviewer's Responses to Questions

**Comments to the Authors:**

Reviewer #1: In this manuscript, Hanke et al. proposed ConNIS, a novel statistical method that accurately identifies essential genes in TraDIS libraries by analytically determining the probability of observing insertion-free sequences within genes, accounting for genome-wide variation in insertion density. It further provides a data-driven criterion for parameter selection to improve comparability across studies and is implemented as an open-source R package and web application for ease of use and reproducibility.

In general, this is a well-written and sound manuscript. The authors presented a robust and innovative method for the identification of essential genes from Transposon Directed Insertion Site Sequencing (TraDIS) data. The methodology is clearly described, the results appear reliable and outperforms existing methods especially under low or medium insertion density, and the conclusions are basically supported by the evidence presented. I believe this work will be of considerable interest to the microbial genomics community.

Major Comments:

1. The demonstrated performance of your method is highly compelling. Given its advantages, the research community would benefit from the systematic re-analysis of existing public TraDIS data. I strongly recommend that you apply your pipeline to all relevant datasets in the DEG database. Generating a unified, high-confidence set of essential gene predictions across multiple organisms and studies using your superior method would be an invaluable resource. This would allow for more accurate comparative genomics and hypothesis generation. I encourage you to make this comprehensive prediction set easily accessible, ideally through a dedicated web portal or a repository, alongside the publication of this manuscript.

2. To further strengthen the manuscript and highlight the biological relevance of your methodological improvements, a more detailed comparative analysis is needed. Specifically, I request a focused section that identifies discrepancies. Please provides a list or table of specific genes for the organisms analyzed where your new predictions of essentiality conflict with the predictions from the original studies. For each major discrepancy, offers a plausible and detailed explanation in biological context. Please explain the gene's function and why the new prediction makes biological sense. This analysis will move beyond simply stating that your method is "better" and will provide concrete, biologically-grounded examples of its value.

3. The binary classification of genes into "essential" and "non-essential" is a known oversimplification. As famously demonstrated by the JCVI-syn3.0 minimal genome project, a third category—quasi-essential genes—is critical for robust growth and fitness. I suggest evaluating the performance of statistical methods on the quasi-essential gene set, and incorporating relevant discussions.

Reviewer #2: Considering that the article “ConNIS and labelling instability: new statistical methods for improving the detection of essential genes in TraDIS libraries” provides a solid methodological innovation, the proposed Consecutive Non-Insertion Sites (ConNIS) method represents a significant advance by offering an analytical solution to calculate the probability of insertion-free sequences—an aspect that previously lacked an exact mathematical foundation.

On the one hand, the study presents a comprehensive experimental design, based on a rigorous comparison with five reference methods using synthetic, semi-synthetic, and real datasets, which strengthens both the validity and generalisability of the results.

On the other hand, the well-structured simulation approach, in which the authors tested 160 parameter combinations and multiple biological scenarios (high and low insertion densities, coldspots, and experimental noise), demonstrates a deep understanding of potential biases in TraDIS data and contributes to a high level of reproducibility and transparency in the analysis.

However, although the mathematical formulation is rigorous, the article may prove challenging to follow for scientists without an advanced statistical background. In addition, some sections are overly technical, which may hinder comprehension for a broader biological audience.

As a future recommendation; acknowledging that it may fall beyond the scope of the present publication—although results are compared with E. coli and Salmonella libraries, the study focuses almost exclusively on statistical performance metrics (MCC, PRC). It would be desirable to include experimental validation, for instance, verification of predicted essential genes through knockout experiments or phenotypic analyses.

Overall, the work is methodologically robust and experimentally convincing, combining an innovative statistical framework with extensive validation. It represents a significant contribution to TraDIS data analysis, particularly under experimental conditions characterised by sparse or uneven insertion densities. Nevertheless, its impact could be further enhanced by including direct experimental validation and by making the presentation more accessible to researchers without a quantitative background.

Reviewer #3: 55: Larivire et al : Lariviere et al.

273: indicated by the rather low precion values. Do you mean precision ?

Figure 3,4,5: There are a lot of similar plots. Could you keep the smallest and largest number of IS/samples , or the most interesting ones, and put the rest in supplemental data maybe ?

Table 1: Do you have a way to compare the MCC you obtained with tuning with those obtained with the previous methods, such as the ones used in the papers cited in the paragraph from line 54 to 69?

**Have the authors made all data and (if applicable) computational code underlying the findings in their manuscript fully available?**

Reviewer #1: Yes

Reviewer #2: Yes

Reviewer #3: Yes

PLOS authors have the option to publish the peer review history of their article (what does this mean? ). If published, this will include your full peer review and any attached files.

**Do you want your identity to be public for this peer review?** For information about this choice, including consent withdrawal, please see our Privacy Policy .

Reviewer #1: No

Reviewer #2: No

Reviewer #3: No

**Figure resubmission:**

**Reproducibility:**



---

## [Decision Letter · Decision Letter 1]

28 Jan 2026

PCOMPBIOL-D-25-01627R1

ConNIS and labeling instability: new statistical methods for improving the detection of essential genes in TraDIS libraries

PLOS Computational Biology

Dear Dr. Hanke,

Thank you for submitting your manuscript to PLOS Computational Biology. After careful consideration, we feel that it has merit but does not fully meet PLOS Computational Biology's publication criteria as it currently stands. Therefore, we invite you to submit a revised version of the manuscript that addresses the points raised during the review process.

We look forward to receiving your revised manuscript.

Kind regards,

Jinyan Li

Academic Editor

PLOS Computational Biology

Ilya Ioshikhes

Section Editor

PLOS Computational Biology

**Reviewers' comments:**

Reviewer's Responses to Questions

**Comments to the Authors:**

Reviewer #1: I am pleased with the authors' substantial revisions, especially the new biological analysis section. However, regarding my first major suggestion, I find the decision to defer all systematic re-analysis to future work to be a missed opportunity that diminishes the potential impact of this otherwise excellent method.

While I appreciate the authors' concerns about data heterogeneity, their argument about data inaccessibility is somewhat overstated. Many high-quality Tn5 TraDIS studies have deposited their raw reads in public repositories. A more impactful and feasible approach would be to include a focused demonstrative analysis. Specifically, I strongly recommend that the authors select a curated set (e.g., 5-10) of publicly available Tn5 TraDIS datasets from the SRA and apply their ConNIS pipeline to generate a unified prediction set.

Providing this curated resource – either as a supplementary table or through a citable data repository – would transform this manuscript from a methods description into a valuable community resource. Given that the authors have already developed a complete, publicly available analysis pipeline, this task should be quite feasible. I believe this addition would significantly enhance the manuscript's utility and should be incorporated prior to publication.

Additionally, I note that the reference list contains formatting problem. Several entries (including refs 4, 8, 9, 10, 11, 12, and 15) are missing the required article numbers. The entire reference list should be carefully reviewed and corrected to meet journal formatting standards.

**Have the authors made all data and (if applicable) computational code underlying the findings in their manuscript fully available?**

Reviewer #1: Yes

PLOS authors have the option to publish the peer review history of their article (what does this mean? ). If published, this will include your full peer review and any attached files.

**Do you want your identity to be public for this peer review?** For information about this choice, including consent withdrawal, please see our Privacy Policy .

Reviewer #1: No

**Figure resubmission:**
---

## [Editor Report · Decision Letter 2]

12 Feb 2026

Dear Mr Hanke,

We are pleased to inform you that your manuscript 'ConNIS and labeling instability: new statistical methods for improving the detection of essential genes in TraDIS libraries' has been provisionally accepted for publication in PLOS Computational Biology.

Best regards,

Jinyan Li

Academic Editor

PLOS Computational Biology

Ilya Ioshikhes

Section Editor

PLOS Computational Biology

---

## [Editor Report · Acceptance letter]

PCOMPBIOL-D-25-01627R2

ConNIS and labeling instability: new statistical methods for improving the detection of essential genes in TraDIS libraries

Dear Dr Hanke,

I am pleased to inform you that your manuscript has been formally accepted for publication in PLOS Computational Biology. Your manuscript is now with our production department and you will be notified of the publication date in due course.

With kind regards,

Anita Estes
